# Tetrodotoxin/Saxitoxins Selectivity of the Euryhaline Freshwater Pufferfish *Dichotomyctere fluviatilis*

**DOI:** 10.3390/toxins13100731

**Published:** 2021-10-16

**Authors:** Hongchen Zhu, Towa Sakai, Yuji Nagashima, Hiroyuki Doi, Tomohiro Takatani, Osamu Arakawa

**Affiliations:** 1Graduate School of Fisheries and Environmental Sciences, Nagasaki University, 1–14 Bunkyo-machi, Nagasaki 852-8521, Japan; zhc957286316@hotmail.com (H.Z.); bb53121017@ms.nagasaki-u.ac.jp (T.S.); taka@nagasaki-u.ac.jp (T.T.); 2Department of Agro-Food Science, Niigata Agro-Food University, 2416 Tainai, Niigata 959-2702, Japan; yuji-nagashima@nafu.ac.jp; 3Nifrel, Osaka Aquarium Kaiyukan, 1–2 Suita, Osaka 565-0826, Japan; doi@kaiyukan.com

**Keywords:** *Dichotomyctere fluviatilis*, pufferfish, saxitoxin (STX), *Takifugu rubripes*, tetrodotoxin (TTX), tissue slice

## Abstract

The present study evaluated differences in the tetrodotoxin (TTX)/saxitoxins (STXs) selectivity between marine and freshwater pufferfish by performing in vivo and in vitro experiments. In the in vivo experiment, artificially reared nontoxic euryhaline freshwater pufferfish *Dichotomyctere fluviatilis* were intrarectally administered a mixture of TTX (24 nmol/fish) and STX (20 nmol/fish). The amount of toxin in the intestine, liver, muscle, gonads, and skin was quantified at 24, 48, and 72 h. STX was detected in the intestine over a long period of time, with some (2.7–6.1% of the given dose) being absorbed into the body and temporarily located in the liver. Very little TTX was retained in the body. In the in vitro experiments, slices of intestine, liver, and skin tissue prepared from artificially reared nontoxic *D. fluviatilis* and the marine pufferfish *Takifugu rubripes* were incubated in buffer containing TTX and STXs (20 nmol/mL each) for up to 24 or 72 h, and the amount of toxin taken up in the tissue was quantified over time. In contrast to *T. rubripes*, the intestine, liver, and skin tissues of *D. fluviatilis* selectively took up only STXs. These findings indicate that the TTX/STXs selectivity differs between freshwater and marine pufferfish.

## 1. Introduction

Tetrodotoxin (TTX), a potent neurotoxin, is the main toxin component of marine pufferfish of the genus *Takifugu*, including *Takifugu rubripes* and *Takifugu pardalis*, which inhabit the coastal waters of Japan [1,2,3]. Organs such as the liver, ovary, and, in some species, the skin, are generally highly toxic, while the muscle and testes are nontoxic or only weakly toxic [2,4,5]. While the exact source of the TTX in pufferfish is unclear, it is thought to accumulate via the food chain starting with TTX-producing bacteria [2]. Saxitoxins (STXs) are the main toxin group found in freshwater pufferfish, such as *Pao* and *Leiodon* (both formerly known as *Tetraodon*), inhabiting the coastal waters of Southeast Asian countries [6,7,8,9,10]. In these freshwater pufferfish, high concentrations of the toxin accumulate in the skin and ovary, whereas relatively low concentrations are usually detected in the liver [8,9,10,11]. STXs are a group of neurotoxins associated with the toxification of bivalves [12]. STX, which is the best-known component of STXs, shares a similar molecular weight, toxicity, and intoxication mechanism as TTX [12]. While freshwater pufferfish generally contain only STXs, some marine species, such as *Sphoeroides* pufferfish from Florida, *Arothron* and *Canthigaster* pufferfish from the Philippines, Japanese coastal waters, or the Caribbean Sea contain both TTX and STXs [13,14,15,16,17]. Like TTX, STXs in these pufferfish are thought to accumulate from exogenous sources, i.e., the food chain that starts with STX-producing dinoflagellates in marine environments and STX-producing cyanobacteria in freshwater environments [7,13,15,16,18,19].

The TTX/STXs that accumulate in pufferfish are from exogenous sources, and therefore, pufferfish that are artificially reared from hatching with nontoxic feed do not contain TTX/STXs [20]. Several in vivo toxin administration experiments have been performed with such nontoxic cultured individuals [9,21,22,23,24,25,26]. TTX administration experiments in marine *Takifugu* pufferfish [21,22,23,24,25] demonstrated unique TTX kinetics in pufferfish from the intestine to the liver, as well as from the liver to the skin/ovary, which may differ between sexes and change as the fish grow and/or mature. Following intramuscular administration of TTX and decarbamoylsaxitoxin (dcSTX) to the freshwater pufferfish *Pao turgidus*, only dcSTX was transferred and accumulated in the skin [9]. Selective accumulation of TTX or STXs was observed in the marine species *T. pardalis* and freshwater species *Pao suvattii* administered both TTX and STXs into the intestine by oral gavage. *T. pardalis*, which naturally contains TTX, showed selective accumulation of TTX, and *P. suvattii*, which naturally contains STXs, showed selective accumulation of STXs [26]. These findings suggest that the TTX/STXs ratio in pufferfish depends more strongly on the inherent toxin selectivity of the pufferfish than the pervasiveness of TTX/STXs in their environment. Thus, in vivo toxin administration experiments allow us to evaluate the selective toxin accumulation ability of pufferfish, which may reflect the toxin profile of wild individuals to some extent.

Experiments by Nagashima et al. [27,28] and Matsumoto et al. [29,30] using an in vitro tissue slice incubation method revealed that the liver of marine *Takifugu* pufferfish, unlike nontoxic general marine fish such as parrot-bass and greenling, takes up TTX, while it barely takes up STXs like nontoxic general marine fish. These results clearly reflect the toxin profile of wild pufferfish or the toxin accumulation ability evaluated in the in vivo toxin administration experiments. Gao et al. [31] applied the liver tissue slice incubation method developed by Nagashima et al. to skin and intestine tissues and showed that TTX uptake was similarly high in the skin, intestine, and liver of *T. rubripes*. The STXs uptake ability of the skin and intestine, and the TTX/STXs uptake ability of freshwater pufferfish tissues, however, remain to be clarified.

Pufferfish of the genus *Dichotomyctere* (formerly known as *Tetraodon*) are distributed in South and Southeast Asia, but their habitat varies depending on the species; some inhabit brackish water, and others live from brackish water to freshwater [32,33,34,35,36,37,38,39,40,41]. Among *Dichotomyctere* pufferfish, *Dichotomyctere nigroviridis* and *Dichotomyctere ocellatus* (also known as *D. steindachneri*), which are found in brackish water in Thailand, possess TTX as the main toxin like marine *Takifugu* pufferfish [42,43,44]. On the other hand, *D. fluviatilis*, the target species of this study, is a euryhaline freshwater species that is also found in brackish water, but, unlike *D. nigroviridis* and *D. ocellatus*, is widely distributed in freshwater ecosystems such as the Ganges River and rivers around the Mekong Delta, Bengal, and Yangon [32,33,36,38,39,40,41]. To our knowledge, the toxicity and toxin profile of *D. fluviatilis* have not yet been reported. Therefore, we sought to elucidate the differences in toxin selectivity between freshwater and marine pufferfish by first conducting an in vivo TTX/STX administration experiment using artificially reared nontoxic *D. fluviatilis* individuals (Figure 1), and then evaluating the TTX/STXs accumulation ability at the individual level. An in vitro tissue slice incubation method was applied to investigate the TTX/STXs uptake ability of the intestine, liver, and skin tissues at the tissue level, compared with that of *T. rubripes*.

## 2. Results

### 2.1. In Vivo Toxin Administration Experiment with D. fluviatilis

In the in vivo toxin administration experiment, artificially reared nontoxic *D. fluviatilis* individuals were intrarectally administered feed homogenates containing TTX (24 nmol/fish) and STX (20 nmol/fish), and the toxin content in the intestine, liver, muscle, gonads, and skin was quantified by instrumental analyses after 24, 48, and 72 h. The results are shown in Figure 2 as the change in the relative toxin amount (% of the given dose) in each tissue. As for STX, 35.5% of the given dose was retained in the pufferfish body at 24 h, which decreased to 29.1% at 48 h and 14.3% at 72 h. A part of the STX retained in the body (2.7–6.1% of the given dose) was absorbed from the intestine and transferred to the liver, but the majority remained in the intestine. No transfer to other tissues (muscle, gonads, and skin) was observed. In contrast, only a tiny amount of TTX (0.02–0.12%) remained in the intestine, and no TTX was detected in any other tissues throughout the 72 h period. The amount of STX and TTX in the intestine was significantly different at all rearing times evaluated (*t*-test, *p* < 0.05).

### 2.2. In Vitro Tissue Slice Incubation Experiment with D. fluviatilis and T. rubripes

In the in vitro tissue slice incubation experiment, intestine, liver, and skin tissue slices were prepared from artificially reared nontoxic *D. fluviatilis* and *T. rubripes*, and incubated with buffer containing TTX (20 nmol/mL) and STXs (neosaxitoxin (neoSTX), 10 nmol/mL; dcSTX, 1 nmol/mL; STX, 9 nmol/mL) for *D. fluviatilis*; and TTX (20 nmol/mL) and STX (20 nmol/mL) for *T. rubripes* up to 24 or 72 h, and the amount of toxin taken up by each tissue slice over time was quantified (Figure 3). All of the *D. fluviatilis* tissues took up much more STXs than TTX. The STXs content in the intestine tissue increased rapidly within 1 h, reached a maximum (11.0 nmol/g) at 8 h, and then decreased slightly. The STXs content in the liver was lower than that in the other tissues (5.2 nmol/g at 24 h), but the STXs content in the skin increased rapidly by 8 h, and then further increased to 12.4 nmol/g at 72 h. In contrast, the TTX content in any tissue did not exceed 1 nmol/g. In all tissues examined, the amounts of STXs and TTX differed significantly (*p* < 0.05) at all incubation times evaluated. In strong contrast to *D. fluviatilis*, the amount of TTX uptake greatly surpassed that of STX in *T. rubripes*. In the intestine and skin, the TTX content increased immediately and reached a maximum (7.9 and 20.8 nmol/g) at 8 h and 48 h, respectively, and then plateaued, whereas, in the liver, the TTX content increased linearly and was highest (31.5 nmol/g) at 24 h. In contrast, the maximum STX content was only 2.3 nmol/g in the intestine, 8.7 nmol/g in the liver, and 4.7 nmol/g in the skin. In the intestine and skin, the amounts of TTX and STX differed significantly (*p* < 0.05) at all incubation times evaluated except at 72 h in the skin.

Changes in the molar ratio of each STXs component taken up by *D. fluviatilis* tissue slices are shown in Figure 4. The ratio of neoSTX, dcSTX, and STX in the STXs added to the incubation buffer was 50%, 5%, and 45%, respectively, but in the STXs taken up in the tissue slices, the ratio of neoSTX (16.1–43.8%) was lower and the ratio of STX (50.6–81.9%) was generally higher than the initial ratios in all tissues. No obvious change in the ratio of dcSTX (2.0–6.3%) was observed.

## 3. Discussion

Our findings revealed that when TTX/STX was administered into the intestine of the euryhaline freshwater species *D. fluviatilis*, only STX remained in the intestine, and some was absorbed into the body and temporarily transferred to the liver. In contrast to the marine species *T. rubripes*, the intestine, liver, and skin tissues of *D. fluviatilis* selectively took up only STXs in the presence of both TTX and STXs. These findings indicated that *D. fluviatilis* and *T. rubripes* have completely opposite TTX/STXs selectivity, not only at the individual level but also at the tissue level.

### 3.1. Selective Toxin Accumulation Ability of D. fluviatilis

In the in vivo toxin administration experiment, a part of the STX administered to the intestine of *D. fluviatilis* remained in the intestine, even after 72 h, and during this period, a small amount of the STX that was absorbed into the body was transferred to only the liver. In contrast, when dcSTX was administered intramuscularly to the freshwater pufferfish *P. turgidus*, more than 90% of the dcSTX remaining in the body was eventually transferred to the skin where it accumulated [9]. Toxins administered directly into the muscle would more easily transfer to the skin adjacent to the muscle, whereas toxins administered directly into the intestine would presumably be less readily absorbed into the body. In *P. suvattii* which were administered TTX/STX to the intestine by oral gavage, however, some STX was observed in the intestine, but a greater proportion of STX was absorbed into the body and transferred mainly to the ovary and skin, where it accumulated [26]. While oral gavage and intrarectal routes of administration differ with respect to the location of the intubation (mouth or anus, respectively), the amount of toxin administered into the intestine was not significantly different. Therefore, *D. fluviatilis*, like other freshwater pufferfish, tends to retain STXs in its body, but its ability to accumulate STXs is probably lower than that of *P. suvattii*. In marine pufferfish, a large amount of TTX is transferred to the ovary as sexual maturation progresses [24,45,46]. Freshwater pufferfish generally possess large amounts of STXs in the ovary as well, and therefore, because the *D. fluviatilis* specimens used in the present study were immature (gonadosomatic index was less than 0.4), the STXs may not have transferred to the ovaries.

TTX was only detected in the areas where the toxin was administered and not in other areas, as in the case of *P. turgidus* [9] and *P. suvattii* [26]. TTX is not absorbed into the body from the intestine, and would thus be discharged with feces, or if absorbed, it would quickly decompose or be excreted, but it would not accumulate. In contrast, in marine *Takifugu* pufferfish, TTX administered into the muscle or intestine is transferred first to the liver, where it accumulates, and then to the skin or ovary via the bloodstream [21,22,23,25]. *D. fluviatilis*, like other freshwater pufferfish, has little TTX accumulation ability in contrast to *Takifugu* marine pufferfish.

### 3.2. Selective Toxin Uptake Ability of The Tissues in D. fluviatilis and T. rubripes

The in vitro tissue slice incubation experiment clearly indicated that, in contrast to *T. rubripes*, the intestine, liver, and skin tissues of *D. fluviatilis* selectively take up only STXs. The skin showed the highest maximum STXs uptake, followed by the intestine, whereas in the liver it was relatively low and comparable to that in *T. rubripes*. This result is consistent with the general intra-body toxin distribution in wild freshwater pufferfish (high toxicity in the skin and relatively low toxicity in the liver) [8,9,10,11] but contradicts the result of the in vivo toxin administration experiments (STX absorbed into the body was transferred only to the liver and was not detected in the skin). The reason for this is unclear at this time but will be clarified in the process of elucidating the molecular mechanisms involved in the selective toxin accumulation ability of individuals and the selective toxin uptake ability of specific tissues.

In *T. rubripes*, the liver, intestine, and skin tissues take up TTX remarkably, while the liver takes up little STXs [27,29,30,31]. The present study additionally demonstrated that the intestine and skin also take up little STXs. In comparison with the STXs uptake of *D. fluviatilis*, the higher TTX uptake of *T. rubripes*, especially in the liver, was noticeable. This result is consistent with the fact that the liver of the marine *Takifugu* pufferfish is generally highly toxic [2,4,5]. Thus, it can be inferred that the toxin profile and the intra-body toxin distribution of wild pufferfish depend to some extent on the selective toxin uptake ability of each tissue.

In the tissue slice incubation experiment of *D. fluviatilis*, STXs taken up by the tissue slices tended to comprise higher percentages of STX in general compared with the STXs added to the incubation buffer. This finding suggests that tissues such as the intestine, liver, and skin preferentially take up STX among the STXs components. On the other hand, the reduction of *O*22-sulfate and *N*1-hydroxyl groups inside the body of bivalves and xanthid crabs results in the conversion of gonyautoxins and neoSTX to STX [47,48,49]. The STXs harbored by wild freshwater pufferfish usually comprise mainly STX, with neoSTX and dcSTX as minor components [6,8,9,10], but it is unclear whether such toxin composition merely reflects the toxin composition of toxic prey organisms, or whether the selectivity involves toxin uptake by pufferfish tissues and/or internal conversion of components. This requires further investigation.

### 3.3. Toxin Profile and Ecology of D. fluviatilis in Nature 

Little information is available on the toxicity or toxin profile of wild *D. fluviatilis*. Untario et al. [50] reported that TTX in the crude liver extract of *D. fluviatilis* increased the intracellular calcium level of HeLa cells and induced apoptosis, but the identification of TTX was insufficient and the presence or absence of STXs in the crude extract was not investigated. The results of in vivo toxin administration and in vitro tissue slice incubation experiments in the present study suggest that *D. fluviatilis* is most likely a weakly toxic species that possesses STXs to some extent, but barely possesses TTX. In contrast, *D. nigroviridis* and *D. ocellatus*, members of the same genus, contain high concentrations of TTX, mainly in the skin [42,43,44]. In 2013, Kottelat [32] divided the former genus *Tetraodon* into 4 genera; *Tetraodon* (7 species), *Pao* (13 species), *Leiodon* (1 species), and *Dichotomyctere* (6 species). Of these, the genera *Pao* and *Leiodon* are found in freshwater ecosystems such as rivers, lakes, and streams in South and Southeast Asia [32], and harbor STXs mainly in their skin and ovary [6,7,8,9,10]. *D. fluviatilis* is a euryhaline freshwater species that inhabits brackish water as well as freshwater, whereas *D. nigroviridis* and *D. ocellatus* are found in marine and brackish waters, such as coastal and estuarine areas in Thailand, northeastern Indonesia, and the Malay Peninsula, and are more like marine species [32,37,41]. Assuming that the original TTX transportation/accumulation system was altered to transport and accumulate STXs when some marine pufferfish moved into freshwater, the genus *Dichotomyctere* may be in the transitional stage of such an alteration, in which there is a mixture of species living from marine to brackish waters and accumulating TTX and a mixture of species living from brackish water to freshwater and acquiring weak STXs accumulation ability. In this study, we were not able to obtain wild individuals of *D. fluviatilis*, whose toxicity and toxin profile require further clarification.

### 3.4. Future Perspective

Although TTX/STXs selectivity in marine and freshwater pufferfish differs at the individual and tissue levels, the underlying mechanisms involved remain to be resolved at the molecular level. In marine pufferfish, pufferfish STX- and TTX-binding protein (PSTBP) [51] is presumed to be involved in the absorption, transportation, and accumulation of TTX [52]. On the other hand, in our recent study on the phylogeny and toxin profile of Cambodian freshwater pufferfish, the STXs accumulation ability differed between the 2 species with different sequences of tributyltin-binding protein type 2 (TBT-bp2) [53], suggesting that TBT-bp2 is involved in STXs accumulation in freshwater pufferfish [10]. In future studies, we will gather enough data on the distribution of PSTBP/TBT-bp2 isoforms in marine and freshwater pufferfish and their expression kinetics in each tissue, as well as information on the individual TTX/STXs accumulation ability and tissue TTX/STXs uptake ability. By comparing and analyzing these accumulation abilities, we may be able to infer the molecular mechanisms and underlying processes of TTX/STXs accumulation in pufferfish. These additional studies are in progress.

## 4. Conclusions

In the present study, we conducted an in vivo toxin administration experiment and in vitro tissue slice incubation experiment to evaluate differences between the TTX/STXs selectivities in marine and freshwater pufferfish and revealed that the euryhaline freshwater pufferfish *D. fluviatilis*, like the pure freshwater pufferfish *P. suvattii*, which possess STXs in nature, is endowed with the STXs-selective toxin accumulation ability at the individual level and the STXs-selective toxin uptake ability at the tissue level. Although the STXs accumulation ability might be lower than that of *P. suvattii* and *D. fluviatilis*, in contrast to marine *Takifugu* pufferfish, which mainly possess TTX in nature, had little TTX accumulation ability or TTX uptake ability in the tissues. The molecular mechanisms underlying differences in the TTX/STXs selectivity of marine and freshwater pufferfish remain unclear. We are currently collecting data on PSTBP and TBT-bp2 as a target to further elucidate the molecular mechanisms and evolutionary processes underlying TTX/STXs accumulation in pufferfish.

## 5. Materials and Methods

### 5.1. Pufferfish Specimens

Of 10 artificially reared nontoxic *D. fluviatilis* (body length, 7.3 ± 0.9 cm; body weight 24.1 ± 9.5 g), nine were used for the in vivo toxin administration experiment, and one was used for the in vitro tissue slice incubation experiment together with a single nontoxic cultured *T. rubripes* (body length, 28.5 cm; body weight 750 g).

### 5.2. Toxin Preparation

TTX (purity >60%) was prepared from the ovaries of *T. pardalis*, and STX (purity >80%) and a mixture of neoSTX, dcSTX, and STX were prepared from the xanthid crab *Zosimus aeneus* according to the previously reported method [54,55]. TTX and STX were used for the toxin administration experiment and tissue slice incubation experiment for *T. rubripes*. Because a sufficient amount of STX could not be obtained, a mixture of neoSTX, dcSTX, and STX was used with TTX for the tissue slice incubation experiment for *D. fluviatilis*.

### 5.3. In Vivo Toxin Administration Experiment

The toxin administration experiment was basically conducted using the method described by Gao et al. [26]. A mixed aqueous solution of TTX and STX was added at a ratio of 2:1 (v:w) to Otohime C2 artificial feed for marine juvenile fish (Marubeni Nisshin Feed Co. Ltd., Tokyo, Japan), and homogenized to prepare a feed homogenate containing TTX (240 nmol/mL) and STX (200 nmol/ mL). In a preliminary experiment, when the homogenate was administered by oral gavage, the *D. fluviatilis* test fish spit it out. Therefore, in the full experiment, the homogenate was administered intrarectally at a dose of 0.1 mL/fish to the test fish, which were equally divided into three aerated 30 L tanks (25 °C) filled with dechlorinated tap water. Three individual fish were randomly collected at 24, 48, and 72 h after toxin administration, and the intestine, liver, muscle, gonads, and skin were collected. Each tissue type was homogenized with 0.1% acetic acid and heated in a boiling water bath for 10 min. The supernatant was centrifuged at 2300 *g* for 15 min and passed through an HLC-DISK membrane filter (0.45 µm, Kanto Chemical Co. Inc., Tokyo, Japan) before submitting to toxin quantification as described below.

### 5.4. In Vitro Tissue Slice Incubation Experiment

The tissue slice incubation experiment was performed using the method described by Gao et al. [31]. Briefly, circular tissue slices (8 mm diameter, ~1 mm thick) were prepared from the intestine, liver, and dorsal skin of *D. fluviatilis* and *T. rubripes* test fish. Each slice was incubated with 1.5 mL incubation buffer (160 mM NaCl, 4.8 mM KCl, 23.8 mM NaHCO_3_, 0.96 mM KH_2_PO_4_, 1.5 mM CaCl_2_, 1.2 mM MgSO_4_, 12.5 mM HEPES, and 5.0 mM D-glucose; adjusted to pH 7.4 with NaOH solution) containing TTX (20 nmol/mL) and STXs (neoSTX, 10 nmol/mL; dcSTX, 1 nmol/mL; STX, 9 nmol/mL) for *D. fluviatilis* and TTX (20 nmol/mL) and STX (20 nmol/mL) for *T. rubripes* in a 15 mL plastic tube aerated with O_2_ and CO_2_ at a 9:1 ratio at 20 °C for a maximum of 24 or 72 h. During the incubation, three slices of each tissue were collected at 20 min, and at 1, 8, and 24 h for the intestine; 8 and 24 h for the liver; and 8, 24, 48, and 72 h for the skin, and weighed after washing with neutral phosphate buffer (0.15 M NaCl and 0.01 M Na_2_HPO_4_; adjusted to pH 7.0 with 0.15 M NaCl and 0.01 M NaH_2_PO_4_). Then, 1 mL of 0.1% acetic acid was added to each slice, and the slices were disrupted by ultrasonication and heated in a boiling water bath for 10 min. After centrifugation at 20,000 *g* for 15 min, the supernatant was passed through an HLC-DISK membrane filter (0.45 µm, Kanto Chemical Co. Inc., Tokyo, Japan), and then used for toxin quantification as described below.

### 5.5. Toxin Quantification

Liquid chromatography-tandem mass spectrometry and high-performance liquid chromatography with post-column fluorescence derivatization were used to quantify TTX and STXs, respectively, as described previously [17]. For TTX quantification, an Alliance 2690 Separations Module (Waters, Milford, MA, USA) with a Mightysil RP-18 GP column (2.0 × 250 mm, particle size 5 µm, Kanto Chemical Co. Inc., Tokyo, Japan) was used for the chromatography with mobile phase (30 mM heptafluorobutyric acid in 1 mM ammonium acetate buffer (pH 5.0)) at a 0.2 mL/min flow rate. The eluate was introduced into a Quattro micro^TM^ API detector (Waters, Milford, MA, USA) in which the TTX was ionized by positive-mode electrospray ionization with a desolvation temperature of 400 °C, a source block temperature of 120 °C, and a cone voltage of 40 V, and monitored at *m/z* 162 (for quantitative analysis) and 302 (for qualitative analysis) as product ions (collision voltage 38 V) with *m/z* 320 as a precursor ion through a MassLynx^TM^ NT operating system (Waters, Milford, MA, USA).

For STXs quantification, chromatographic separation was performed using Prominence Ultra-Fast Liquid Chromatography (Shimadzu, Kyoto, Japan) with an LiChroCART Superspher RP18(e) column (4.0 × 250 mm, particle size 4 µm, Kanto Chemical Co. Inc., Tokyo, Japan). STX, neoSTX, and dcSTX were separated using a mobile phase of 2 mM heptanesulfonic acid in 4% acetonitrile-30 mM ammonium phosphate buffer (pH 7.3) at a flow rate of 0.8 mL/min. The column temperature was set at 35 °C. The eluate from the column was mixed continuously with 50 mM periodic acid and 0.2 M KOH containing 1 M ammonium formate and 50% formamide, and heated at 65 °C. The formation of fluorophores was monitored at 392 nm with 336-nm excitation through a RF-20A XS Prominence Fluorescence Detector (Shimadzu, Kyoto, Japan).

The limit of detection and limit of quantification of TTX was 0.0009 nmol/mL (*S/N* = 3) and 0.003 nmol/mL (*S/N* = 10), and those of the STXs were 0.001–0.007 nmol/mL and 0.003–0.02 nmol/mL, respectively.

### 5.6. Data Analyses

In the in vivo toxin administration experiment, the TTX/STX amount per individual tissue was calculated from the tissue weight and the TTX/STX content per gram tissue and expressed as a relative value to the given dose. The mean and standard deviation (SD) of three individuals at each rearing time were then calculated to compare the amount of TTX and STX retained in the test fish. For the intestine, each numerical value (x) was converted to arcsine (x/100), and then the Student *t*-test was used to test the significance of differences in the amounts of STX and TTX at each rearing time. In the in vitro tissue slice incubation experiment, the mean and SD of the TTX/STXs content per gram tissue of three tissue slices at each incubation time were calculated to compare the amount of TTX and STXs taken up by the tissue slices. The Student *t*-test was used to test the difference between the amounts of TTX and STXs at each incubation time.

## Figures and Tables

**Figure 1 toxins-13-00731-f001:**
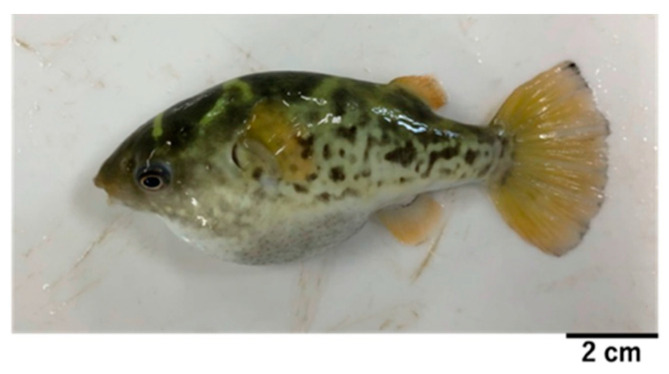
*Dichotomyctere fluviatilis*.

**Figure 2 toxins-13-00731-f002:**
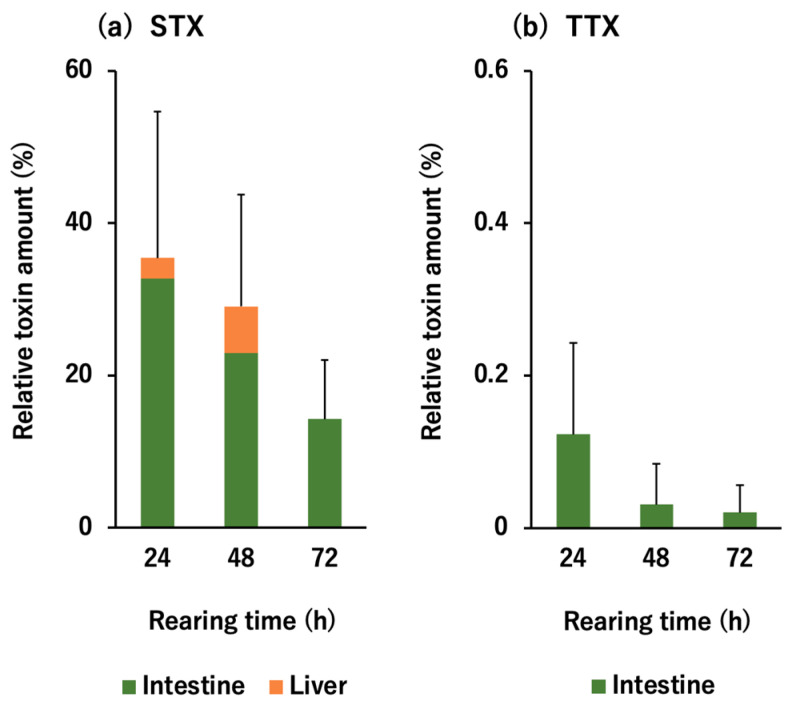
Relative tetrodotoxin (TTX) (**a**) and saxitoxin (STX) (**b**) amount (% of the administered dose) in each tissue of *D. fluviatilis* at 24, 48, and 72 h after TTX/STX administration. Data are shown as means (columns) and standard deviations (SDs) (error bars) of 3 individuals. Note that the scales on the vertical axis differ between STX and TTX by a factor of 100.

**Figure 3 toxins-13-00731-f003:**
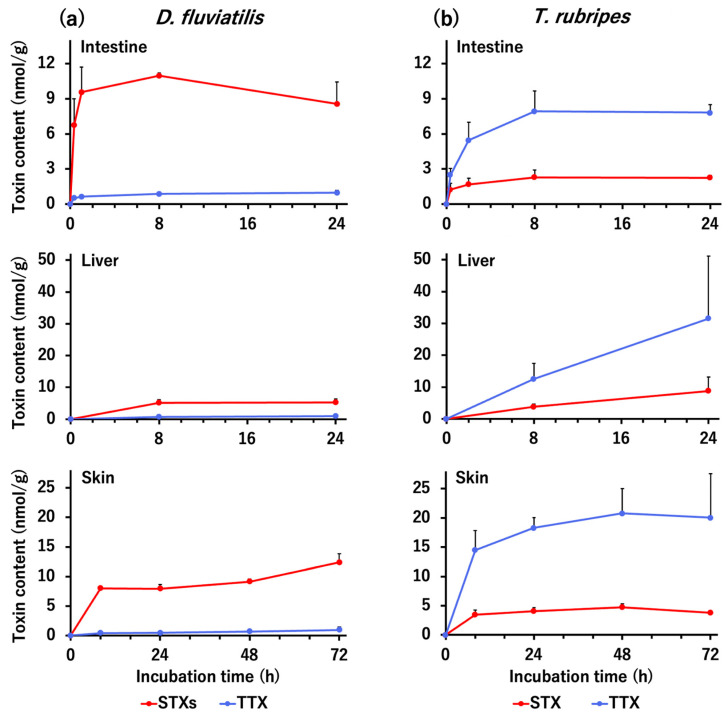
Changes in toxin content in the intestine, liver, and skin slices of *D. fluviatilis* (**a**) and *T. rubripes* (**b**) during incubation for up to 24 or 72 h. Each slice was incubated with buffer containing TTX (20 nmol/mL) and STXs (neosaxitoxin (neoSTX), 10 nmol/mL; decarbamoylsaxitoxin (dcSTX), 1 nmol/mL; STX, 9 nmol/mL) for *D. fluviatilis* and TTX (20 nmol/mL) and STX (20 nmol/mL) for *T. rubripes*. Data are shown as means (dots) and SDs (error bars) of 3 slices.

**Figure 4 toxins-13-00731-f004:**
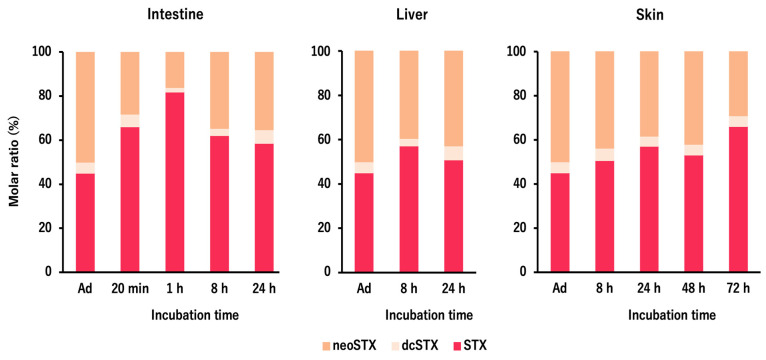
Changes in the molar ratio of each STXs component taken up by each tissue of *D. fluviatilis* during incubation for 24 or 72 h. Ad: STXs added to the incubation buffer.

## Data Availability

Not applicable.

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
