# Peer review of "Tetrodotoxin/Saxitoxins Selectivity of the Euryhaline Freshwater Pufferfish Dichotomyctere fluviatilis"

_toxins, 2021, doi:10.3390/toxins13100731_

Round 1

Reviewer 1 Report

The article entitled “Tetrodotoxin/Saxitoxins Selectivity of the Euryhaline Freshwater Pufferfish Dichotomyctere fluviatilis” explores the toxin retention capability of two species of pufferfish (freshwater and marine). This is a very fascinating topic since the toxicity pathways in most species of pufferfishes are still poorly understood. Therefore, I consider the article suitable for publication in Toxins as it contributes to a better understanding of trends in STX/TTX accumulation in pufferfish. I have only a few remarks, listed below.

Title: as you also studied another species Takifugu rubripes (though only in vitro) I think you should include it in the title or at least as a keyword.

Abstract: before reading the article the abstract is quite difficult to understand. I think it will be clearer if you explain from the beginning that you performed an in vivo experiment with D. fluviatilis and another in vitro experiment with D. fluviatilis and T. rubripes and then you state your results and conclusions. Also, this phrase: “When tissue slices of intestine, liver, and skin prepared from artificially reared nontoxic individuals of D. fluviatilis and the marine pufferfish Takifugu rubripes were incubated in incubation buffer containing TTX and STXs (20 nmol/mL each) for up to 24 or 72 h, and the amount of toxin taken up in each tissue slice was quantified over time.” (lines 11-14) is too long and seems uncompleted. You may substitute “When” by “Then”.  Furthermore, you mention in material and methods that you performed the experiment as in Gao et al. 2019. They administered the toxin by oral gavage, so I am not sure if the word “intrarrectally” is correct here and throughout the document. Please check it.

Key contribution: delete the semicolon. Mention just the species you studied, since you cannot generalize your experiments to all freshwater and marine pufferfish.

Figure 1. D. fluviatilis  

Results:

L90. “Intrarrectally” please check it is correct.

Figure 2. Since you did not detect STX in muscle, gonads or skin and you mentioned it in the text I think it is not necessary to display them in the plot. The same for TTX. If you want to show the small amount of TTX, then you can change the y-axis to be in the range of TTX values, otherwise, the graph does not add any relevant information.

Discussion:

I think the discussion needs to be arranged in a clearer way, may be it will help the addition of subsections, for example: 3.1. In vivo experiments in D. fluviatilis, 3.2. In vitro experiments in D. fluviatilis and T. rubripes and 3.3. In vivo vs. in vitro experiments results in D. fluviatilis. It is just a suggestion.

L 140-142. “…indicated for the first time that freshwater and marine pufferfish have completely opposite TTX/STXs selectivity, not only at the individual level but also at the tissue level”. This phase seems too enthusiastic. Other previous studies suggested freshwater and marine pufferfish had different toxin selectivities and you only tested two species, so it is quite a generalization. 

L190 “..weakly toxic species that possesses STXs, and barely possesses TTX”. Well, STX is a very potent toxin as well. The fact that the fish do not accumulate TTX does not necessarily means that is less toxic. 

L226 “accumulate information” does not sound well to me. Maybe “we will gather enough data” “we will have enough information”. Try to avoid repetition of “accumulation” so many times. Also, you may want to cite your ongoing work as (xxx, in preparation).

Material and methods:

4.2. Toxin preparation. This section is quite hard to understand: You obtained pure TTX and STX plus a mix of STX-derivatives. And then you performed the in vivo experiment with the pure toxins (D. fluviatilis). The in vitro experiment was then performed with pure TTX along with the mix of STXs for D. fluviatilis but pure STX for T. rubripes.   Why did not you use the mix of STX for both in vitro experiments? Do you think the differences you found in in vivo and in vitro experiments with D. fluviatilis could be related to the fact that you added these derivatives in the in vitro which were not present in in vivo?

I think you can add a section on data analyses.

Author Response

Thank you very much for your valuable comments. We revised our manuscript according to the comments, as indicated below (revised parts are indicated in red font).

The article entitled “Tetrodotoxin/Saxitoxins Selectivity of the Euryhaline Freshwater Pufferfish Dichotomyctere fluviatilis” explores the toxin retention capability of two species of pufferfish (freshwater and marine). This is a very fascinating topic since the toxicity pathways in most species of pufferfishes are still poorly understood. Therefore, I consider the article suitable for publication in Toxins as it contributes to a better understanding of trends in STX/TTX accumulation in pufferfish. I have only a few remarks, listed below.

Title: as you also studied another species Takifugu rubripes (though only in vitro) I think you should include it in the title or at least as a keyword.

Takifugu rubripes” is now included as a keyword in the revised manuscript (L17).

Abstract: before reading the article the abstract is quite difficult to understand. I think it will be clearer if you explain from the beginning that you performed an in vivo experiment with D. fluviatilis and another in vitro experiment with D. fluviatilis and T. rubripes and then you state your results and conclusions. Also, this phrase: “When tissue slices of intestine, liver, and skin prepared from artificially reared nontoxic individuals of D. fluviatilis and the marine pufferfish Takifugu rubripes were incubated in incubation buffer containing TTX and STXs (20 nmol/mL each) for up to 24 or 72 h, and the amount of toxin taken up in each tissue slice was quantified over time.” (lines 11-14) is too long and seems uncompleted. You may substitute “When” by “Then”. 

The abstract has been revised according to the comments (L4-7, 11).

Furthermore, you mention in material and methods that you performed the experiment as in Gao et al. 2019. They administered the toxin by oral gavage, so I am not sure if the word “intrarectally” is correct here and throughout the document. Please check it.

The word “intrarectally” is correct. The relevant part of the Materials and Method section has been revised (L288-290).

Key contribution: delete the semicolon. Mention just the species you studied, since you cannot generalize your experiments to all freshwater and marine pufferfish.

The semicolon has been deleted and the species are now mentioned in the revised manuscript (L20-21).

Figure 1. D. fluviatilis 

“An artificially reared individual of” has been deleted (L90).

Results:

L90. “Intrarrectally” please check it is correct.

As mentioned above.

Figure 2. Since you did not detect STX in muscle, gonads or skin and you mentioned it in the text I think it is not necessary to display them in the plot. The same for TTX. If you want to show the small amount of TTX, then you can change the y-axis to be in the range of TTX values, otherwise, the graph does not add any relevant information.

Figure 2 are now modified according to the comments, and a supplementary sentence was added to the caption (L109-110).

Discussion:

I think the discussion needs to be arranged in a clearer way, may be it will help the addition of subsections, for example: 3.1. In vivo experiments in D. fluviatilis, 3.2. In vitro experiments in D. fluviatilis and T. rubripes and 3.3. In vivo vs. in vitro experiments results in D. fluviatilis. It is just a suggestion.

Based on the comments, the Discussion section has been re-arranged into subsections. The Results section has also been divided into subsections accordingly.

L 140-142. “…indicated for the first time that freshwater and marine pufferfish have completely opposite TTX/STXs selectivity, not only at the individual level but also at the tissue level”. This phase seems too enthusiastic. Other previous studies suggested freshwater and marine pufferfish had different toxin selectivities and you only tested two species, so it is quite a generalization.

“for the first time” is now deleted and “freshwater and marine pufferfish” is changed to “D. fluviatilis and T. rubripes” in the revised manuscript (L153-155).

L190 “..weakly toxic species that possesses STXs, and barely possesses TTX”. Well, STX is a very potent toxin as well. The fact that the fish do not accumulate TTX does not necessarily means that is less toxic.

We presume the species to be weakly toxic not because it does not accumulate TTX but because we are discussing from the in vivoexperiment results that “D. fluviatilis, like other freshwater pufferfish, tends to retain STXs in its body, but its ability to accumulate STXs is probably lower than that of P. suvattii” (L170-172). To avoid misunderstanding, “to some extent” has been added after “that possesses STXs” in the revised manuscript (L223).

L226 “accumulate information” does not sound well to me. Maybe “we will gather enough data” “we will have enough information”. Try to avoid repetition of “accumulation” so many times.

“we will accumulate information” has been changed to “we will gather enough data” (L249).

Also, you may want to cite your ongoing work as (xxx, in preparation).

We are currently collecting data from a wide variety of pufferfish species, but we are not yet ready to prepare a paper.

Material and methods:

4.2. Toxin preparation. This section is quite hard to understand: You obtained pure TTX and STX plus a mix of STX-derivatives. And then you performed the in vivo experiment with the pure toxins (D. fluviatilis). The in vitro experiment was then performed with pure TTX along with the mix of STXs for D. fluviatilis but pure STX for T. rubripes. Why did not you use the mix of STX for both in vitro experiments? Do you think the differences you found in in vivo and in vitro experiments with D. fluviatilis could be related to the fact that you added these derivatives in the in vitro which were not present in in vivo?

Initially, we were planning to use STX for all experiments, but as we could not prepare a sufficient amount of STX, we used a mixture of STXs instead. Indeed, it would have been easier to understand to use a mixture of STXs for both of the in vitro experiments, or to use STX for the in vitro experiment with D. fluviatilis and a mixture of STXs for the in vitro experiment with T. rubripes. However, about half of the STXs mixture is STX. Although it may have some effect, I think it is unlikely that the differences observed between the in vivo and in vitroexperiments with D. fluviatilis are fundamentally due to the addition of these derivatives in the in vitro which were not present in in vivo. Therefore, we believe that this setting does not diminish the value of this study.

I think you can add a section on data analyses.

A section on data analyses is now added in the revised manuscript (L343-353). Accordingly, statistics-related statements have been added to the Results section (L104-105, 122-124, 129-131).

Based on the journal’s request, many other minor revisions have been made throughout the manuscript to reduce the ‘similarity rate’ of the manuscript (the revised parts are not shown).

Reviewer 2 Report

The manuscript is a well-written report dealing with documenting the selectivity of the euryhaline species Dichotomyctere fluviatilis to retain tetrodotoxin or saxitoxin in their bodies, investigated at tissue level, in comparison with the marine species Takifugu rubripes. It does add valuable new information in the field, especially with regard to the differences in TTX/STX selectivity between freshwater and marine species. There are only few minor points, which could be improved to increase the manuscript’s reader-friendliness and correct some minor mistakes, but also the potential ethical concerns of the study need to be addressed, as indicated below:

Points for Revision: 

1. Introduction:
- Page 1, lines 28-29: “Pufferfish do not biosynthesize TTX, but accumulate it through the food chain that starts with TTX-producing bacteria”: There is still some controversy on the exact sources of TTX in the scientific community, so this should be rephrased to reflect that this is not yet completely clarified.
- Page 2, lines 59-61: please rephrase this sentence, it should be clarified what it means “like general marine fish”. What is the rule in that case?

2. Results
- Page 3, lines 101-104 (Figure 2): The right panel of the figure should be magnified (or given separately in a sub-panel with a different scale) to display the results for TTX. There are minor columns shown, which would need magnification to become visible for the readers.
- Page 4, lines 122-124: Please indicate in the figure caption the amount of TTX & STX contained in the incubation medium (for each species). The figure should be self-explanatory.

4. Materials and Methods:
- Please provide details on animal experimentation licensing, in order to address the ethical concerns of the study – fish are vertebrates. 
- Please explain the rationale for choosing intrarectal administration in this work, in contrast to previous similar works where oral gavage or intramuscular administration was used

Author Response

Thank you very much for your valuable comments. We revised our manuscript according to the comments, as indicated below (revised parts are indicated in red font).

The manuscript is a well-written report dealing with documenting the selectivity of the euryhaline species Dichotomyctere fluviatilis to retain tetrodotoxin or saxitoxin in their bodies, investigated at tissue level, in comparison with the marine species Takifugu rubripes. It does add valuable new information in the field, especially with regard to the differences in TTX/STX selectivity between freshwater and marine species. There are only few minor points, which could be improved to increase the manuscript’s reader-friendliness and correct some minor mistakes, but also the potential ethical concerns of the study need to be addressed, as indicated below:

 Points for Revision:

  1. Introduction:

- Page 1, lines 28-29: “Pufferfish do not biosynthesize TTX, but accumulate it through the food chain that starts with TTX-producing bacteria”: There is still some controversy on the exact sources of TTX in the scientific community, so this should be rephrased to reflect that this is not yet completely clarified.

The sentence has been rephrased according to the comments (L29-30).

- Page 2, lines 59-61: please rephrase this sentence, it should be clarified what it means “like general marine fish”. What is the rule in that case?

 The sentence has been rephrased according to the comments (L64-65).

  1. Results

- Page 3, lines 101-104 (Figure 2): The right panel of the figure should be magnified (or given separately in a sub-panel with a different scale) to display the results for TTX. There are minor columns shown, which would need magnification to become visible for the readers.

The right panel of the figure 2 are now magnified in the revised manuscript, and a supplementary sentence was added to the caption (L109-110).

- Page 4, lines 122-124: Please indicate in the figure caption the amount of TTX & STX contained in the incubation medium (for each species). The figure should be self-explanatory.

The amount of TTX and STXs contained in the incubation medium are now indicated in the figure caption in the revised manuscript (L134-137).

  1. Materials and Methods:

- Please provide details on animal experimentation licensing, in order to address the ethical concerns of the study – fish are vertebrates.

We essentially conduct animal experiments following the Guidelines for Animal Experimentation of Nagasaki University and the Regulations of the Animal Care and Use Committee, Nagasaki University. These guidelines and regulations, however, currently do not include fish, amphibians, and invertebrates.

- Please explain the rationale for choosing intrarectal administration in this work, in contrast to previous similar works where oral gavage or intramuscular administration was used.

The rationale for choosing intrarectal administration are now explained in the revised manuscript (L288-290).

Based on the journal’s request, many other minor revisions have been made throughout the manuscript to reduce the ‘similarity rate’ of the manuscript (the revised parts are not shown).

Reviewer 3 Report

Manuscript entitled „Tetrodotoxin/Saxitoxins Selectivity of the Euryhaline Freshwater Pufferfish Dichotomyctere fluviatilis” is an interesting, well-written and well-planned experimental work, providing new data, showing differences in the distribution and accumulation capacity of TTX/STX in this species of freshwater pufferfish. I fully support the publication of this manuscript; however I recommend the minor revision of manuscript. Small corrections should be made to the text.

Introduction

line 36 - according to the context of this sentence and the earlier information given in the introduction, it seems better that the sentence should look like this:

In general, marine pufferfish contain no STXs, but some marine species, such as Sphoeroides pufferfish from Florida, Arothron and Canthigaster pufferfish from the Philippines, Japanese coastal waters, or the Caribbean Sea have both TTX and STXs [13–17].

line 45 - please add examples of references at the end of the sentence after individuals

line 51 – after TTX/STXs should be were instead was

Results

line 107 - explain the abbreviation neoSTX with the full name

Figure 2 - abbreviations should be defined the first time they appear in figure: TTX/STX; SD

Figure 4 - abbreviations should be defined the first time they appear in figure: neoSTX, dcSTX

Additionally, in the Results section in the description of Figure 2 and 3 you wrote that the obtained data are presented as mean and standard deviation. Please explain what statistical methods were used to calculate these results as there is no information on this in the Materials and methods section. A description of statistical analysis of the obtained data should be found in this section of the manuscript.

Discussion

line 152 - Please explain in more detail, based on the literature, why D. fluviatilis accumulates much smaller amounts of STX compared to P. turgidus and P. suvattii, for example in the context of different routes of administration of the toxin in the compared experiments.

line 152 - move the sentence from line 152 to line 156 to the part of the text below describing no TTX accumulation

Conclusion - at the end of the manuscript there should be a summary containing the conclusions from the conducted research and an explanation for what purpose of described research was carried out and, above all, what is its scientific significance.

Author Response

Thank you very much for your valuable comments. We revised our manuscript according to the comments, as indicated below (revised parts are indicated in red font).

Manuscript entitled „Tetrodotoxin/Saxitoxins Selectivity of the Euryhaline Freshwater Pufferfish Dichotomyctere fluviatilis” is an interesting, well-written and well-planned experimental work, providing new data, showing differences in the distribution and accumulation capacity of TTX/STX in this species of freshwater pufferfish. I fully support the publication of this manuscript; however I recommend the minor revision of manuscript. Small corrections should be made to the text.

Introduction 

line 36 - according to the context of this sentence and the earlier information given in the introduction, it seems better that the sentence should look like this:

In general, marine pufferfish contain no STXs, but some marine species, such as Sphoeroides pufferfish from Florida, Arothron and Canthigaster pufferfish from the Philippines, Japanese coastal waters, or the Caribbean Sea have both TTX and STXs [13–17].

The story up to this sentence can be summarized as follows;
Marine pufferfish of the genus Takifugu mainly have TTX.
Freshwater pufferfish of the genera Pao and Leiodon have STXs instead of TTX.
I think the sentence that follows should be "Freshwater pufferfish contain no TTX (freshwater pufferfish contain only STXs), but some marine pufferfish have both TTX and STXs", not "marine pufferfish contain no STXs, but some marine pufferfish have both TTX and STXs" as you suggested.

"freshwater pufferfish contain no TTX" are now changed to "freshwater pufferfish contain only STXs” in the revised manuscript (L38).

line 45 - please add examples of references at the end of the sentence after individuals

Reference has been added as suggested (L47).

line 51 – after TTX/STXs should be were instead was

The relevant sentence has been rewritten to reduce the ‘similarity rate’ of the manuscript (L52-54).

Results

line 107 - explain the abbreviation neoSTX with the full name

“neoSTX” are now explained with the full name in the revised manuscript (L114). Similarly, “dcSTX” has been described with the full name (L51).

Figure 2 - abbreviations should be defined the first time they appear in figure: TTX/STX; SD

Revised as suggested (L108-109).

Figure 4 - abbreviations should be defined the first time they appear in figure: neoSTX, dcSTX

The caption of Figure 3 has been revised, where neoSTX and dcSTX are defined (L134-137).

Additionally, in the Results section in the description of Figure 2 and 3 you wrote that the obtained data are presented as mean and standard deviation. Please explain what statistical methods were used to calculate these results as there is no information on this in the Materials and methods section. A description of statistical analysis of the obtained data should be found in this section of the manuscript.

A section on data analyses is now added in the revised manuscript (L343-353). Accordingly, statistics-related statements have been added to the Results section (L104-105, 122-124, 129-131).

Discussion

line 152 - Please explain in more detail, based on the literature, why D. fluviatilis accumulates much smaller amounts of STX compared to P. turgidus and P. suvattii, for example in the context of different routes of administration of the toxin in the compared experiments.

Based on the comments, a discussion on the administration routes has been added (L162-172).

line 152 - move the sentence from line 152 to line 156 to the part of the text below describing no TTX accumulation

Some words are now added in the revised manuscript, as the purpose of the sentence will change a bit if it is moved to the next paragraph (L172-176).

Conclusion - at the end of the manuscript there should be a summary containing the conclusions from the conducted research and an explanation for what purpose of described research was carried out and, above all, what is its scientific significance.

Conclusions section is now added in the revised manuscript (L256-268).

Based on the journal’s request, many other minor revisions have been made throughout the manuscript to reduce the ‘similarity rate’ of the manuscript (the revised parts are not shown).